# Mediterranean Diet and Lung Function in Adults Current Smokers: A Cross-Sectional Analysis in the MEDISTAR Project

**DOI:** 10.3390/nu15051272

**Published:** 2023-03-03

**Authors:** Roxana-Elena Catalin, Francisco Martin-Lujan, Patricia Salamanca-Gonzalez, Meritxell Palleja-Millan, Felipe Villalobos, Antoni Santigosa-Ayala, Anna Pedret, Rosa M. Valls-Zamora, Rosa Sola

**Affiliations:** 1Research Support Unit Camp of Tarragona, Department of Primary Care Camp de Tarragona, Institut Català de la Salut, 43202 Reus, Spain; 2CENIT Research Group, Fundació Institut Universitari Per a la Recerca a l’Atenció Primària de Salut Jordi Gol i Gurina (IDIAPJGol), 08007 Barcelona, Spain; 3Departament of Medicine and Surgery, Facultat de Medicina i Ciències de La Salut, Universitat Rovira i Virgili, 43201 Reus, Spain; 4Functional Nutrition, Oxidation and Cardiovascular Disease Group (NFOC-SALUT), Universitat Rovira i Virgili, 43201 Reus, Spain

**Keywords:** Mediterranean Diet, lung function, smoking, primary care centres

## Abstract

Background: Previous studies have shown that adherence to the Mediterranean Diet (MeDi) has a positive impact on lung function in subjects with lung disease. In subjects free of respiratory diseases, but at risk, this association is not yet well established. Methods: Based on the reference data from the MEDISTAR clinical trial (Mediterranean Diet and Smoking in Tarragona and Reus; ISRCTN 03.362.372), an observational study was conducted with 403 middle-aged smokers without lung disease, treated at 20 centres of primary care in Tarragona (Catalonia, Spain). The degree of MeDi adherence was evaluated according to a 14-item questionnaire, and adherence was defined in three groups (low, medium, and high). Lung function were assessed by forced spirometry. Logistic regression and linear regression models were used to analyse the association between adherence to the MeDi and the presence of ventilatory defects. Results: Globally, the pulmonary alteration prevalence (impaired FEV1 and/or FVC) was 28.8%, although it was lower in participants with medium and high adherence to the MeDi, compared to those with a low score (24.2% and 27.4% vs. 38.5%, *p* = 0.004). Logistic regression models showed a significant and independent association between medium and high adherence to the MeDi and the presence of altered lung patterns (OR 0.467 [95%CI 0.266, 0.820] and 0.552 [95%CI 0.313, 0.973], respectively). Conclusions: MeDi adherence is inversely associated with the risk impaired lung function. These results indicate that healthy diet behaviours can be modifiable risk factors to protect lung function and reinforce the possibility of a nutritional intervention to increase adherence to MeDi, in addition to promoting smoking cessation.

## 1. Introduction

The etiopathogenesis of chronic respiratory diseases, such as asthma and chronic obstructive pulmonary disease (COPD), is multifactorial [1,2]. It includes some individual conditions that cannot be modified, such as age, gender or genetic predisposition and some environmental factors, whereas lifestyle factors, including smoking and environmental exposure, physical activity and diet are modifiable and can be changed [3]. Although cigarette smoking is the predominant risk factor, there is consistent evidence from epidemiologic studies that other environmental factors are also involved in chronic airflow limitation, including outdoor and indoor air pollution or exposure to biomass fuel, and second-hand smoke during pregnancy or early childhood [4,5]. Lower socioeconomic status has been consistently associated with airflow obstruction, but it is unclear whether this pattern reflects environmental exposures and infections, poor nutrition and unhealthy dietary habits, physical inactivity, or other related factors [6].

Dietary intake may be a major risk factor for impaired lung function, and healthy dietary habits may protect respiratory health [3]. Cross-sectional and prospective evidence has shown that certain natural antioxidants and fatty acids provided from foods could neutralise the harmful effect of tobacco on lung function. Thus, for example, the consumption of fruits and vegetables, with a high content of antioxidant vitamins, phenolic compounds, minerals and dietary fibre, as well as to omega 3 fatty acids, found in oily fish and seafood, show benefits on the pathophysiology of the respiratory disease [7]. In contrast, a high consumption of processed meat has been associated with worse lung function and a higher risk of diseases of the respiratory system, which could be explained by its high content of nitrites, nitrates and advanced glycation products that cause inflammation and oxidative damage [8]. In addition, excessive alcohol intake has also been found to have detrimental effects on lung function [9].

Although an individual analysis of dietary components has been evaluated, an approach focused on investigating individual nutrients or foods could overlook the complexity of their interactive or synergistic effects within the diet, as neither foods nor nutrients are consumed in isolation [10,11]. Thus, the study of dietary patterns, characterised by relatively high consumption of some foods with relatively low consumption of others in a single exposure, is a more attractive approach to study the association of diet and respiratory diseases [12]. Previous research has shown that the adherence to a “healthy” (or prudent) dietary pattern (high in fruits, vegetables, whole grains, lean meat, fish, antioxidants, and fibre and low in fat and dairy products) may reduce the risk of lung impairment [3]; conversely, a “western” type pattern (high in processed and refined foods, high in sugars and fats, and low in antioxidant and fibre levels) can increase the risk of COPD and asthma attacks in children and adults [13,14].

To date, the most researched and highly promising dietary pattern for exerting a protective effect on respiratory function is the Mediterranean Diet (MeDi) [3]. The traditional MeDi is well-balanced diet, characterised by high consumption of vegetables, legumes, fruits, fish, nuts, wholegrains including non-refined cereals, and olive oil, foods rich in antioxidants, phenolic compounds and mono- and polyunsaturated fatty acids, with beneficial effects on inflammation and oxidation [15]. This dietary pattern has been the subject of research for years and the evidence for its beneficial health effects is overwhelming [16], particularly regarding cardiovascular prevention [17]. In this context, the interest of extending the MeDi recommendation to tobacco-related respiratory disease is evident. However, there is still limited evidence regarding the MeDi pattern and lung function in healthy populations but with sufficient risk factors for disease. Therefore, the main objective of the present study was to study the association between MeDi adherence and the risk of impaired lung function in a cohort of smokers without known respiratory disease.

## 2. Materials and Methods

### 2.1. Study Design

Observational study based on data obtained in the recruitment phase of the MEDISTAR study, a parallel, multicentre, cluster-randomised, controlled clinical trial to assess the effect of MeDi on lung function in smokers without previous respiratory disease (identifier Clinicaltrials.org: NCT03362372, accessed on 1 December 2017). Details of the study design have been reported elsewhere [18].

### 2.2. Selection of Participants and Obtaining the Sample

The study population was obtained from patients attended in 20 primary care centres that provide medical care to a population of about 280,000 inhabitants, managed by the Catalan Institute of Health in Tarragona, Catalonia, Spain.

The inclusion criteria in the MEDISTAR clinical trial were: age 25 to 75 years and active smoker together with cumulative consumption ≥ 10 pack-years. Subjects with a history of respiratory disease, chronic or terminal disorder, or any reason that might alter follow-up or testing during the study were excluded. All participants signed an informed consent before being included.

Figure 1 illustrates the CONSORT flowchart of participants from MEDISTAR study to those included in the present analysis.

A total of 403 subjects were enrolled between 1 July 2017 and 30 June 2018 and were randomly assigned into two groups: an intervention group to increase MeDi adherence through nutritional re-education, and a control group that follows their usual eating style. For the present analysis, the participants of both groups were stratified by their degree of adherence to the MeDi according to the 14-item MEDAS questionnaire that was developed in the PREDIMED trial [17]. Responses to this questionnaire have previously been shown to be valid for assessing adherence to the MeDi [19]. Low adherence to the MD was considered when the score obtained in the test was 0–6 points, medium adherence when it was 7–8 points, and high adherence when it was 9–14 points.

### 2.3. Study Variables

The main study variable was the presence of pulmonary alteration. Pulmonary function examinations were performed by trained and certified technicians in agreement with the American Thoracic Society and European Respiratory recommendations [20]. Forced spirometry was done using a spirometer model DATOSPIR-600© (SIBELMED, Barcelona, Spain) with a disposable Lilly type transducer, and the accuracy of the device was verified daily. The spirometric parameters were measured as a percentage of the predicted values, focusing on the lung function parameters FEV1 (forced expiratory volume in the first second), FVC (forced vital capacity) and the relationship between them (FEV1/FVC ratio). Spirometry was done before and after inhalation of a bronchodilator. For the present analysis, we only considered pre-bronchodilator parameters.

Abnormal lung function was defined as an FVC and/or FEV1 value < 80% of the predicted value or FEV1/FVC ratio < 0.7. In addition, three spirometric patterns were considered: “normal spirometry” (FVC and FEV1 ≥ 80% of the predicted value and FEV1/FVC ratio ≥ 0.7), “obstructive ventilatory defect” (FEV1/FVC ratio < 0.7), and “non-obstructive ventilatory defect” (FVC < 80% of the predicted value and FEV1/FVC ratio > 0.7) [20].

As secondary variables were considered the following data:

(1) Socio-demographic: age, sex, marital status (classified as single/widowed, married and divorced/separated), educational level (does not know to write or read, primary studies, secondary studies, and university degree) and employment status (working, working at home, unemployed/disability/retired, and student).

(2) Clinical morbidities: medical history of cardiovascular disease, circulatory disease, oncologic disease and endocrinology disease.

(3) Smoking: current consumption (number cigarettes per day) and cumulative consumption (pack-years smoked, calculated by multiplying the number of packs of cigarettes smoked per day by the number of years the person has smoked).

(4) Alcohol consumption: grams of alcohol/week. To unify criteria when calculating alcohol consumption, the World Health Organization stipulated its measurement through the Standard Drink Unit (SDU). In Spain, one SDU equals 10 g of alcohol, and the consumption limit is <20 g/day (2 SDUs) for men and <10 g/day (1 SDU) for women, and the risk consumption > 28 SDUs/week for men and >14–17 SDUs/week for women, assuming that any consumption (however minimal) implies risk [21].

(5) Physical activity: according to the short Catalan version of the International Physical Activity Questionnaire (IPAQ-SF), the participants were classified as engaged in high (vigorous physical activity of at least 1500 Mets, 3 or more days per week, or a combination of walking and/or moderate to vigorous physical activity, 7 days per week), moderate (moderate physical activity of at least 600 Mets and/or walk for at least 30 min, 5 or more days per week) and low physical activity (some physical activity but it is insufficient), and inactive (no physical activity) [22].

(6) Physical examination: blood pressure (systolic [SBP] and diastolic [DBP], measured twice in sitting position on the right arm, calculated as the mean value of the two measurements, in mmHg), height (cm, using a conventional stadiometer), weight (kg, with the participant in light clothing and without shoes), BMI (in kg/m^2^) and its categorisation according to the classification criteria of the World Health Organization (normal weight, <25.0 kg/m^2^; overweight, 25–29.9 kg/m^2^ and obesity, ≥30 kg/m^2^), and waist circumference (WC; measured midway between the lower rib and the iliac crest with a standard anthropometric tape).

(7) Laboratory data: levels of glucose (mg/dL), total cholesterol and fractions, triglycerides (mg/dL), and basic blood count parameters (haemoglobin [g/dL], haematocrit [%], erythrocytes [10^6^/mm^3^]).

### 2.4. Statistical Analysis

Data were extracted from a centralised database created ad hoc for the MEDISTAR study. For this analysis, participants were classified into three groups according to the degree of adherence to the MeDi: low (MeDi-low), medium (MeDi-medium) or high (MeDi-high).

At first we made a descriptive analysis differentiated by the 3 adherence groups. The categorical variables were described by their frequency distribution and continuous variables were described by the mean and standard deviation (SD) or median, first quartile and third quartile, depending on whether or not they had a normal distribution, respectively. The Shapiro-Wilk test was used to decide the normality of the variables with a significance of 0.01. To detect differences between three groups, χ^2^ test was used for categorical variables and ANOVA-test or Kruskall-Wallis test was employed according to the normal distribution of each variable. For post-hoc comparisons Tukey or Benjamini and Hochberg tests were used.

In order to analyse the relationships between MeDi adherence and lung function outcomes, multiple logistic and linear regression models were applied. Multivariate logistic regression models were performed for three outcomes, impaired FEV1, impaired FVC and impaired FEV1 and/or FVC. For each response variable, three types of models are shown according to the adjustment variables: an unadjusted model with the 3 groups of adherence to the MeDi as the only explanatory variable; an adjusted model adding sex and age, and finally an adjusted model adding multiple variables selected according to a stepwise algorithm performed in both directions. The model shown in the results was chosen according to the minimal value of Akaike Information Criterion (AIC) and clinical relevance. The results are presented as odds ratio (OR) with 95% confidence intervals (CI). Multiple linear regression models were applied to explain the MeDi adherence group and impaired FEV1% predicted value and impaired FVC% predicted value. For each of the two response variables, three models are shown according to the adjustment variables: an unadjusted model with the 3 groups of adherence to the MeDi as the only explanatory variable; an adjusted model adding sex and age, and finally an adjusted model adding multiple variables selected according to a stepwise algorithm performed in both directions, the model shown in the results was chosen according to the minimal value of AIC and clinical relevance. Results are presented using the beta coefficients with 95%CI.

All statistical analysis tests will be conducted with R Statistics package (R foundation for statistical computing, Vienna, Austria; version 4.1.2) and will be considered significant when *p* < 0.05.

### 2.5. Ethical Approval

The protocol was approved by the Ethics Committee of the University Institute for Primary Care Research—IDIAPJGol (registration code P17/089). The study was conducted according to the principles established by the Helsinki Declaration, in the Good Clinical Practice guidelines of the International Conference on Harmonisation (ICH GCP), and Spanish legislation regarding the protection of personal information was also followed. The subjects received information about the objectives of the study and the activities related to their participation, and signed an informed consent before their inclusion. ClinicalTrials.gov Identifier: NCT03362372.

## 3. Results

A total of 403 participants were included with a mean age of 51.1 (SD 10.1) years, 66% women. Regarding lung function parameters, the study population had a mean %FEV1-predicted value of 92.6 (SD 17.4), a mean %FVC-predicted value of 90.3 (±16.0), and a mean FEV1/FVC ratio of 0.78 (SD 0.74). Regarding the degree of adherence to the MeDi the prevalence of the low adherence was 23.8% (95% CI 19.9 to 28.2), the medium adherence was 40.0% (95% CI 35.3 to 44.8) and the high adherence was 36.2% (95% CI 31.7 to 41.0).

Table 1 summarises the main general characteristics of participants grouped according to levels of adherence to the MeDi. As can be seen, there were no significant differences between the groups in terms of distribution by sex, age, level of physical activity, alcohol consumption and smoking. We can see that in the high adherence to MeDi group there is a tendency towards a higher alcohol consumption which can be explained by the fact that wine is considered part of the MeDi and the MEDAS questionnaire does not discriminate between wine and other beverages. It should be noted that the wine is included in the MeDi. In addition, no statistically significant differences were observed between groups regarding the physical examination and the analytical variables studied, except for a more favourable lipid profile in those with high adherence (significant value for LDL cholesterol, and close for total cholesterol and triglycerides). The Appendix A shows the positive answers given by the participants and the influence of each question of the MEDAS questionnaire on the total score, according to the degree of adherence to the MeDi (Appendix A).

Table 2 shows the results of the main parameters for evaluating lung function and the prevalence of impairment according to the degree of adherence to the MeDi. The prevalence of altered lung function was 28.8% overall (impaired FEV1 and/or FVC), 26.1% for FVC and 21.1% for FEV1, but it differed according to the degree of adherence to the MeDi. The prevalence of pulmonary alteration was significantly higher in the low-adherence group compared to the medium and high-adherence group. (38.5% vs. 24.2% and 27.4%, respectively; *p* = 0.044). Regarding the type of spirometric pattern, the differences did not reach statistical significance.

The logistic regression models for the independent variables related to the impaired pulmonary function are shown in Table 3. We carried out an unadjusted model (only included the MEDI adherence), a model adjusted by sex and age, and a third one adding other variables that were selected according to the minimal value of AIC criterion and their clinical relevance (BMI and smoking current consumption). Logistic regression analysis showed a significant relationship between the presence of impaired lung function and the degree of adherence to the MeDi: regarding the subjects with low adherence, those with medium and high adherence presented a lower probability of functional alteration (OR 0.510 [95%CI from 0.416 to 0.946; *p* = 0.016] and 0.602 [95%CI from 0.348 to 1.042; *p* = 0.070], respectively). The degree of adherence to the MeDi remained an independent risk factor after adjusting for sociodemographic, clinical and lifestyle variables (OR 0.467 [95%CI 0.266 to 0.820; *p* = 0.008] and 0.552 [95%CI 0.313 to 0.973; *p* = 0.040], respectively).

The multivariate linear regression models of the independent variables related to the pulmonary function data (FEV1 and FVC) are shown in Table 4. We analysed an unadjusted model (only included the MEDI adherence), a model adjusted by sex and age, and a third one including other variables that were selected according to the minimal value of AIC criterion and its clinical relevance (smoking current consumption, BMI and SDU/week). Considering FEV1 as the main variable, we can observe that the female sex increases the FEV1% predicted value (Beta regression coefficient 5.87; 95%CI 2.242 to 9.497). Regarding tobacco consumption, a higher consumption is inversely related (Beta regression coefficient −0.143; 95%CI from −0.272 to −0.038), while higher alcohol consumption increases the FEV1% predicted value (Beta regression coefficient: 0.374; 95%CI 0.074 to 0.675). Considering the FVC and applying the same models, only tobacco and sex were related with statistical significance to modify the FVC% predicted value (Beta regression coefficient: −0.143 [95%CI −0.239 to −0.048] and 6.029 [%95 CI 2.789 to 9.268], respectively). Regarding the main variable, adherence to the Mediterranean Diet in linear regression was not significantly related to the variation of FEV1% and FVC% predicted value.

## 4. Discussion

This study analyses and reports on the relationship between adherence to the MeDi pattern and respiratory function in a population of adult current smokers free of lung disease, treated at primary care centres in the health area of Tarragona in Catalonia, Spain. Their results show that greater adherence to the MeDi is inversely associated with a lower pulmonary function alteration prevalence (24.2% and 27.4% in medium and high adherence vs. 38.5% in low adherence; *p* = 0.004), and lower risk of presenting impaired lung function (OR 0.467 [95%CI 0.266, 0.820] and 0.552 [95%CI 0.313, 0.973] for medium and high adherence, respectively), after adjusting for potential relevant confounding factors such as smoking, physical activity, and anthropometry. These findings add to the overall evidence for a protective effect of “healthy” dietary pattern on respiratory health and highlight the importance of studying diet as a whole [3].

In general, 2 dietary patterns can be distinguished in nutritional epidemiology: a “healthy” (or prudent) dietary pattern, characterised by high consumption of fruits, vegetables, whole grains, fibre, lean meat and fish, and low in fat and dairy products, and another less healthy “western” type with high intake of refined grains, red and processed meats, chips, fizzy drinks, high in sugars and fats, and low in antioxidant and fibre levels [23]. In a previous cross-sectional study published by our group, we identified three dietary patterns associated with lung function: a Mediterranean-style pattern, a western-style pattern, and an alcohol-consumption pattern [24]. In the adjusted multivariable model, impaired pulmonary function was positively associated with the western-style and alcohol-consumption patterns, but no association was found with the Mediterranean-style diet, especially in women (OR 5.62 [95%CI95 from 1.17 to 27.02], OR 11.4 [95%CI from 2.25 to 58.47], and OR 0.71 [95%CI from 0.28 to 1.79], respectively). Previously, two large prospective studies of US men and women showed that the risk of newly diagnosed COPD decreased as the prudent dietary pattern score increased [25,26]. Similarly, a healthy dietary pattern has been previously described, characterised by high consumption of fruit, vegetables, oily fish and wholemeal cereals, but low consumption of white bread, added sugar, full-fat dairy products, chips and processed meat, which was associated with better lung function and reduced prevalence of COPD among older people in the UK [27]. In contrast, a cross-sectional analysis from the Netherlands found that the “traditional” pattern, was associated with lower lung function and higher COPD prevalence [28]. More recently, a study in a Korean cohort has reported that a dietary pattern low in vegetable intake was negatively associated with lung function (particularly the FEV1/FVC ratio) and a higher prevalence of COPD [29]. The importance of dietary pattern in asthma was also highlighted in two systematic reviews and meta-analyses, concluding that adherence to MeDi may be effective in preventing asthma or wheezing in children; but these associations are controversial in the case of adults [30,31]. Recently, researchers from the ARIC (Atherosclerosis Risk in Communities) study compared the effect of a westernised diet versus a prudent diet on asthma, COPD, respiratory symptoms, and lung function [11]. They found that asthma prevalence was not related to dietary intake pattern, although people who ate a western diet had a higher prevalence of wheeze, cough, and phlegm and lower measures of lung function. In contrast, prudent dietary intake was protective for COPD and cough, as well as lung function deficits.

Diet and nutrition have been recognised as modifiable risk factors for the development and progression of multiple chronic diseases, including lung diseases [32]. Although the fundamental public health message regarding lung diseases continues to be smoking cessation, the multifactorial nature of many chronic lung diseases opens the possibility of intervening in other modifiable risk factors, such as nutrition [33]. Globally, the available evidence shows that a healthy diet lowered the risk of developing COPD, whereas a westernised diet increased the risk [13,34]. The MeDi is a healthy dietary pattern characterised by a high consumption of extra virgin olive oil, fruit, vegetables, fresh produce, nuts and legumes, a low intake of sweetened beverages, red meat and ready-made meals, and a moderate consumption of fish and seafood, poultry, fermented dairy products and red wine (with meals) [15]. Important epidemiological studies have reported the role of the MeDi in the prevention of chronic diseases such as cardiovascular diseases, diabetes or cancer [17,35], but the evidence on its relationship with lung function and the risk of pathology chronic respiratory disease is more limited. A published study using population-based prospective data from the Västerbotten Intervention Program cohort from Sweden, showed that an intermediate and high MeDi score was inversely associated with the development of COPD (after adjustment for smoking intensity, OR 0.73 [95%CI from 0.53 to 0.99] and OR 0.59 [95%CI from 0.35 to 0.97], respectively) [36]. An observational and cross-sectional study conducted among community-dwelling older adults also examined the association between adherence to a MeDi and lung function (evaluated through peak expiratory flow rate; PEF, l/min) [37]. The results of a logistic regression showed a significant association between high adherence to MeDi with reduced risk of having PEF rate < 80% of its peak predictive value (OR 0.65 [95%CI from 0.48 to0.89]). Our study extends these results to a population of current smokers without lung disease, since we observed that those participants with intermediate and high adherence to the MeDi, compared with those with a low score, presented a lower prevalence of lung function alterations during the manoeuvres of spirometry tests (24.2% and 27.4% vs. 38.5%, respectively; *p* = 0.04) and less probability of lung function alteration (OR 0.467 [95%CI from 0.266 to 0.820] and OR 0.552 [95%IC from 0.313 to 0.973], respectively). Overall, the results indicate that adherence to MeDi is an independent predictor of lung function, and dietary interventions could be a possible preventive measure in adults with a high risk of developing impaired lung function. However, caution should be exercised as no intervention studies have been reported so far and, to our knowledge, no direct evidence has been published on the effects of MeDi modification on lung function [38].

The pathophysiological mechanisms to explain the pulmonary benefits associated with MeDi are not yet fully understood [3]. Probably, the most appropriate explanation could be related to anti-inflammatory and antioxidant properties associated with the MeDi pattern [15]. Antioxidants are thought to play a protective role in the pathogenesis of lung impairment by scavenging free radicals and other oxygen species that cause cellular damage and inflammation [39]. Since MeDi protects against cellular oxidation and inflammation in several systems, it is reasonable to consider that these effects would also apply to lung tissues [40]. Various components of this dietary pattern, such as fruit and vegetables, vitamins C and E, flavonoids, ß-carotene, fatty acids, and various minerals have been shown to exert a protective effect on oxidative and inflammatory processes and could have a protective effect on lung function [34,37]. The ECRHS survey, a population-based study, reported that total fruits and vegetables intake is associated with a slower decline in FEV1 and FVC, which might be partly explained by the flavonoid contents in this food group [10]. In the same line, the Health ABC study, a population-based survey in older adults, also showed that a higher intake of antioxidant nutrients was associated with a slower lung function decline [41]. The favourable fatty acid profile of the MeDi, with a high monounsaturated fatty acids and polyunsaturated fatty acids *n*-3, is associated with an anti-inflammatory effect through inhibition of eicosanoids derived from arachidonic acid [42]. MeDi is also accompanied by a high intake of omega-3 fatty acids, through the weekly consumption of fatty fish and shellfish, which provides anti-inflammatory action. Some evidence shows that the consumption of omega-3 fatty acids, mainly eicosapentaenoic acid (C20: 5) and docosahexaenoic acid (C22: 6), interferes with the inflammatory response and can prevent some of the mechanisms involved in the pathophysiology of various diseases [43,44].

Unhealthy lifestyle choices have significant detrimental impacts. Well-established evidence shows that cancer, cardiovascular disease, diabetes, and chronic respiratory disease share modifiable risk factors such as smoking, alcohol consumption, physical inactivity, and nutritional status, as well as unhealthy diet [45]. For this reason, from a public health perspective, interventions to improve lifestyle habits are considered a priority [46].

Tobacco smoke is the most important factor in the etiopathogenesis of respiratory pathology, although other factors may also be involved [1]. In the present study, a sample of current smokers with cumulative consumption ≥ 10 pack-years, aged between 25 and 75 years (both inclusive), was selected. These subjects are the most likely to present impaired lung function associated with smoking [5]. Indeed, both FEV1% predicted value and FVC% predicted value were inversely associated with smoking cumulative consumption (pack-year) in linear regression analysis (β regression coefficient −0.155 [95%CI from −0.272 to −0.038; *p* = 0.010] and −0.143 [95%CI from −0.239 to −0.048; *p* = 0.003], respectively). These new data are in line with previous evidence [4].

Regarding alcohol consumption, in the present study we did not find any independent effect on the alteration of lung function but we did find it with respect to the impaired FEV1% and FVC% predicted value (β regression coefficient 0.374 [95%CI from 0.074 to 0.675; *p* = 0.015] and 0.244 [95%CI from 0.028 to 0.517; *p* = 0.015], respectively). Also in a previous cross-sectional study of our group, impaired pulmonary function was positively associated with alcohol-consumption pattern, especially in women [24]. However, a recent systematic review concludes that the evidence on the influence of alcohol consumption on the rate of decline in lung function and the risk of COPD is still inconsistent [23]. While excessive alcohol consumption is associated with decreased lung function, lower consumption might have protective effects in the general population. Therefore, adequate longitudinal cohort studies are required to clarify the influence of alcohol consumption on the progression of pulmonary function decline.

The finding in our study of no interaction between adherence to the MeDi and physical activity with lung function also deserves comment, since a synergistic effect has been suggested among current smokers. Also, in the ILERVAS cohort study, no synergistic effects were observed between MeDi and physical activity with respect to better lung function [47]. In contrast, a previous study conducted in the Copenhagen City Heart Study suggested the association between physical activity and lung function, but did not consider the influence of diet, which is well known to be closely related to physical activity and nutritional state [48]. In addition, it must be considered that this relationship could be bidirectional, and that lung function could be affecting the physical activity level, since exercise limitation is a well-known consequence of chronic respiratory conditions [49]. However, people with normal lung function (as is the case for the majority of our sample) have physical activity levels in a wide range, and their behaviour is affected by many other factors besides lung function [50]. Additionally, it is important to recognise that people who smoke fewer cigarettes may lead somewhat healthier lifestyles, including diet and regular exercise. Therefore, it is possible that the interaction between smoking, diet, physical activity and lung function can only be adequately studied in clinical studies with samples covering wide ranges of both parameters [51].

Obesity has an essential effect on lung function. Most studies believe that obesity-related indicators are negatively correlated with lung function changes [52]. Obesity interferes with respiratory function due to mechanical compression and chronic inflammation of the airways [53]. This impact on lung function is independent of smoking, although its interaction is likely to potentiate mechanisms of inflammation and lung remodelling [54]. BMI is commonly used clinical measures of central obesity, and their association with lung function has been widely demonstrated [55]. In a previous study, our group demonstrated that worse anthropometry is associated with a greater probability of impaired lung function in smokers without known respiratory disease [56]. Since there is a strong interrelationship between diet and BMI, we adjusted for BMI in the regression models. However, our current study found that greater adherence to the MeDi pattern was associated with better lung function, and that the beneficial effect of diet was not affected by BMI. In this sense, a recent study has reported a higher impact on pulmonary function when metabolic alterations are present with or without obesity [57]. According to their data, the metabolically healthy obese group had better lung function compared to metabolically unhealthy groups. In addition, subjects with a “metabolically healthy” pattern have a lower proportion of cardio-metabolic diseases, such as dyslipidaemia, diabetes or hypertension, which have been associated with decreased lung function [58]. It has also been reported that the relationship between lung function and some adiposity indices present an inverted U-shaped curve, in such a way that the worst values of FEV1 and FVC occur at the extreme values of anthropometry [59]. In the present study, the average BMI was in the overweight range (25~30 kg/m^2^) and the laboratory data could be considered close to the “metabolically healthy” pattern in all three MeDi adherence groups [60]. All of this could explain why we did not observe an independent effect of BMI on lung function. In any case, the extent to which BMI is a confounding factor and/or a mediator of the associations between dietary habits and lung function would require a specific longitudinal study [61].

In this study, we also included adjustment for various sociodemographic factors, since lung function has been correlated with differences in ethnicity, age, and sex, in addition to smoking [62]. Consistent with the findings of other studies, our results also showed gender behaviour with respect to the impact on lung function, but not for age. We observed that women compared to men had a lower risk of impaired lung function (OR 0.51 [95%CI from 0.360 to 0.904; *p* = 0.017]), and FEV1% or FVC% predicted value (β regression coefficient 5.870 [95%CI from 2.242 to 9.497; *p* = 0.002] and 6.220 [95%CI from 2.893 to 9.546; *p* < 0.001], respectively). Sex-associated differences in the effects of increasing adiposity on lung function have been previously reported and values of decreased lung function are expected to be greater in men than women [63]. Furthermore, dietary macronutrients could have different effects on lung function in men and women [64]. However, it is not clear whether the effect of dietary modification may differ between men and women, although sex differences have been described in other studies [65]. In any case, our current data corroborate a different impact on lung function according to sex, reinforcing the relevance of this variable when evaluating lung function associated with the MeDi.

An inverse association between socioeconomic status (educational level, marital status, income, occupation, etc.) and lung function has been described in the epidemiological literature for decades [66]. For this reason we considered the adjustment for socioeconomic factors (family situation, educational level and employment status), even though they may not be relevant from the statistical point of view (according to the AIC index) to be included in the best model of the regression analysis. Most likely, a multitude of confounding factors (specific to each setting) play a role in the complex relationship between socioeconomic factors and lung function. In addition, changes in social factors throughout life could also influence the dietary pattern and the effect of environmental exposures [67].

### Limitations and Strengths

We also recognise several strengths of the present study. Although it is an observational study, it constitutes the first analysis of the association of the MeDi with lung function in the smoking population without previous respiratory disease and, therefore, provides new information that could complement the available evidence. Our group also previously reported on the feasibility of conducting a randomised controlled clinical trial to assess the efficacy of a Social Networks 2.0-supported dietary intervention in primary health care settings [38]. To the best of our knowledge, no other study has been designed to incorporate the dimension of adherence to diet in the smoking patient as a comprehensive form of patient-centred health care [18].

To quantify the main variable, lung function, we use spirometry measurements, which are the gold standard markers for the diagnosis of the most prevalent respiratory diseases [20]. Lung function data were obtained in the context of stringent spirometry protocols with well-trained field workers. In our study, lung damage and respiratory abnormalities were mild in magnitude and even subclinical, but they could have a detrimental impact on long-term health [38]. However, it should be noted that our definition of impaired lung function represents a simplified case definition for epidemiological purposes and not a definitive clinical diagnosis. Additionally, we focus on the MeDi pattern rather than individual nutrients in foods and believe our pattern analysis can provide practical guidance for public health [68].

Besides these strengths, our study also has a number of limitations that are worth noting and which require us to interpret some data with caution. We highlight the cross-sectional design, which prevents us from drawing clear conclusions with respect to causality. Perhaps subjects with higher adherence MeDi may be more health conscious and engage more in healthy behaviours, which could potentially confound associations between diet and lung function. In any case, “reverse causation” does not seem a likely explanation for the main findings, since it is not understood why individuals developing worse lung function would choose to eat a less healthy diet [65]. Although several studies have shown that the dietary pattern in adults remains reasonably stable over time, with the data from this cross-sectional study, we can only assume an association and not a causal relationship between adherence to the MeDi and impaired lung function [69]. In any case, the longitudinal follow-up provided for in this study will help to elucidate the temporal relationships between lifestyle factors, including adherence to the MeDi, and the presence of impaired lung function.

Another limitation of the study is the sample size. We estimated an initial sample size in the MEDISTAR study of 750 volunteers, but we only recruited 403 eligible to participate. The sample size achieved, although comparable to other studies in this field, possibly this limited the statistical power to detect differences between lung function and adherence to the MeDi, and has caused some doubts when interpreting and comparing the results obtained [3]. Despite this, we consider that the database is large enough to allow us to adjust for potential confounding effects of main factors, including sex and age, sociodemographic factors, and factors related to lifestyle such as smoking, consumption of alcohol, physical activity and some nutrition indices, such as the BMI. In this sense, as has been argued, our results are consistent with other previous studies. However, as in any observational study of diet, unaddressed confounding is a concern in interpreting results. Although many known confounding factors were taken into account, the possibility of residual confounding due to other factors that have not been evaluated in our study (such as the other environmental sources of oxidants/antioxidants, air pollution or occupational exposures) cannot be ruled out. In addition, our study population included only middle-aged and older adult smokers, so we also recognise that our study population could represent a group of people who differ from the general population in terms of health awareness and smoking behaviour.

## 5. Conclusions

In conclusion, the results of the present study show that adherence to the MeDi is an independent predictor of impaired lung function in adult smokers without known lung disease. After taking into account other factors related to their sociodemographic characteristics and lifestyle, a medium-high adherence to MeDi diet was associated with a lower risk of impaired lung function. In addition to the preventive benefits of the MeDi for cardiovascular disease, diabetes, and cancer, increased adherence to MeDi pattern could also play a protective role in the pathogenesis of chronic respiratory diseases. Thus, dietary interventions could be a useful preventive measure in adults at high risk, although smoking cessation remains the main target to reduce the burden of these diseases.

## Figures and Tables

**Figure 1 nutrients-15-01272-f001:**
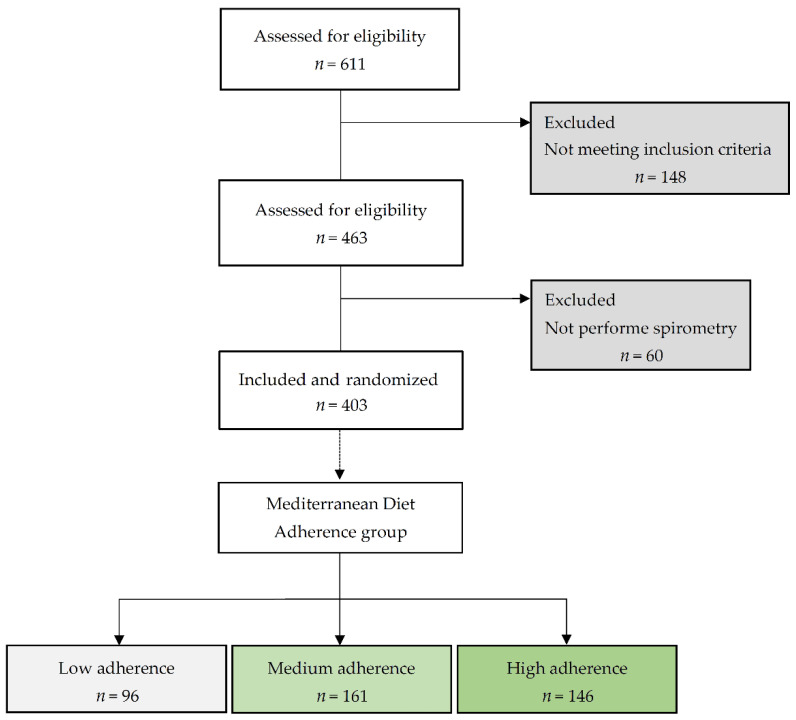
CONSORT diagram adapted for this study: Screening, randomization and initial distribution of the MEDISTAR study participants. Adherence to the Mediterranean Diet was assessed using the 14-item MEDAS validated questionnaire that considers low (0–6 points), medium (7–8 points) and high (9–14 points) adherence [19].

**Table 1 nutrients-15-01272-t001:** Sociodemographic, habits and exploratory descriptive variables according to adherence to Mediterranean Diet group (*).

	Total	Low Adherence	Medium Adherence	High Adherence	*p* Value
	(*n* = 403)	(*n* = 96)	(*n* = 161)	(*n* = 146)	
**Socio-demographics**					
Sex, woman	266 (66.0%)	64 (66.7%)	105 (65.2%)	97 (66.4%)	0.9603
Age	51.1 (10.1)	49.8 (10.4)	51 (10.3)	52.2 (9.66)	0.188
Employment status					0.593
Working	256 (64.2%)	57 (59.4%)	104 (65.4%)	95 (66.0%)	
Unemployed	33 (8.3%)	8 (8.3%)	17 (10.7%)	8 (5.5%)	
Working at home	72 (18.0%)	21 (21.9%)	24 (15.1%)	27 (18.8%)	
Student	38 (9.5%)	10 (10.4%)	14 (8.8%)	14 (9.7%)	
Marital status					0.484
Single	87 (21.6%)	19 (19.8%)	42 (26.1%)	26 (17.8%)	
Married	271 (67.2%)	66 (68.8%)	103 (64.0%)	102 (69.9%)	
Divorced/Windowed	45 (11.2%)	11 (11.5%)	16 (9.9%)	18 (12.3%)	
Education level					0.068
Doesn’t know to read or write	75 (18.7%)	14 (14.6%)	34 (21.1%)	27 (18.6%)	
Primary Studies	142 (35.3%)	45 (46.9%)	51 (31.7%)	46 (31.7%)	
Secondary Studies	141 (35.1%)	28 (29.2%)	53 (32.9%)	60 (41.4%)	
University Studies	44 (10.9%)	9 (9.3%)	23 (14.3%)	12 (8.2%)	
**Health Habits**					
Smoking age, years	17 [15.0; 20.0]	17 [15.0; 20.0]	16 [15.0; 19.0]	16 [15.0; 20.0]	0.548
Current consumption, cigarettes/day	23.0 [15.0; 34.0]	20,3 [12.9; 34.0]	25 [15.0; 34.0]	21.5 [15.0; 34.0]	0.150
Accumulated consumption, packs-year	23.0 [15.0; 34.0]	20.3 [12.9; 34.0]	25 [15.0; 34.0]	21.5 [15.0; 34.0]	0.150
Alcohol consumption,	58.6%	52.1%	55.3%	66.4%	0.048
Alcohol intake, SDU/week	4.88 (6.8)	4.28 (6.7)	4.96 (8.1)	5.11 (5.6)	0.780
Physical activity, IPAQ categorization					0.185
Inactive	33 (8.2%)	10 (10.4%)	12 (7.4%)	11 (7.5%)	
Low	142 (35.2%)	42 (43.8%)	50 (31.1%)	50 (34.2%)	
Moderate	204 (50.6%)	41 (42.7%)	85 (52.8%)	78 (53.5%)	
High	24 (6.0%)	3 (3.1%)	14 (8.7%)	7 (4.8%)	
**Pathological history**					
Cardiovascular disease	18 (4.4%)	6 (6.2%)	6 (3.7%)	6 (4.1%)	0.052
Oncologic disease	25 (6.2%)	12 (12.5%)	8 (4.9%)	5 (3.4%)	0.012
Circulatory disease	59 (14.6%)	20 (20.8%)	24 (14.9%)	15 (10.3%)	0.075
Endocrinological disease	91 (22.6%)	30 (31.2%)	33 (20.5%)	28 (19.2%)	0.064
**Exploratory parameters**					
Weight, kg	70 [61.0; 82.0]	68 [59.0; 80.2]	69 [59.0; 83.0]	72 [63.0; 83.0]	0.222
BMI, Kg/m^2^	27.0 (5.1)	26.9 (4.6)	26.6 (5.3)	27.6 (5.2)	0.187
WC, cm	90.9 (15.3)	91.1 (14.5)	89.2 (14.8)	92.5 (16.2)	0.244
SBP, mmHg	125 (15.3)	123 (14.5)	124 (14.7)	127 (16.4)	0.147
DBP, mmHg	77.5 (9.4)	76.7 (10.3)	77.5 (8.8)	78 (9.6)	0.658
**Laboratory data**					
Glucose, mg/dL	87.3 [80.0; 102.0]	86.0 [80.0; 102.0]	88.3 [80.0; 106.0]	87.4 [81.0; 99.0]	0.512
Total cholesterol, mg/dL	213 (45.6)	208 (41.8)	219 (47.5)	209 (45.4)	0.088
LDL Cholesterol, mg/dL	127 (35.4)	124 (35.8)	133 (34.5)	123 (35.6)	0.035
HDL Cholesterol, mg/dL	61.8 (17.6)	60.8 (16.0)	62 (18.0)	62.1 (18.2)	0.835
Triglycerides, mg/dL	128 [98.0; 179.0]	118 [91.5; 156.0]	146 [105.0; 189.0]	126 [99.9; 173.0]	0.070
Haemoglobin, g/dL	14.0 [12.9; 14.8]	14.2 [12.9; 14.8]	14.2 [13.0; 14.8]	13.8 [12.9; 14.8]	0.532
Haematocrit, %	42.1 [39.2; 44.7]	42.3 [39.2; 45.2]	42.3 [39.7; 44.8]	41.8 [39.0; 44.0]	0.168
Erythrocytes, 10^6^/mm^3^	4.4 [4.2; 4.6]	4.4 [4.2; 4.6]	4.4 [4.3:4.6]	4.4 [4.2; 4.5]	0.173

(*) Adherence to the Mediterranean Diet was assessed using the 14-item MEDAS validated questionnaire that considers low (0–6 points), medium (7–8 points) and high (9–14 points) adherence [19]. Data are presented with frequencies (%), or the mean (standard deviation) or as median [quartile 1; quartile 3], according to the type of variable and if they have a normal distribution or not, respectively. The Shapiro-Wilk test was used to decide the normality of the variables with a significance of 0.01. To detect differences between groups, χ^2^ test was used for categorical variables and ANOVA-test or Kruskall-Wallis test was employed according of the normal distribution of each variable. For post-hoc comparisons Tukey or Benjamini and Hochberg were used. SDU: standard drink unit (an SDU equals 10 g of alcohol); BMI: body mass index; WC: waist circumference; DBP: diastolic blood pressure; SBP: systolic blood pressure; LDL: low-density lipoprotein; HDL: high-density lipoprotein.

**Table 2 nutrients-15-01272-t002:** Lung function according to the degree of the Mediterranean Diet Adherence (*).

	Total	Low Adherence	Medium Adherence	High Adherence	*p* Value
	(*n* = 403)	(*n* = 96)	(*n* = 161)	(*n* = 146)	
**Pulmonary values**					
FEV1, L	2.7 (0.7)	2.7 (0.7)	2.7 (0.7)	2.7 (0.7)	0.982
FEV1, %predicted	92.6 (17.4)	90.8 (17.6)	93.7 (17.7)	92.6 (17.0)	0.433
FVC, L	3.5 (0.9)	3.5 (0.9)	3.5 (0.8)	3.51 (0.9)	0.973
FVC, %predicted	90.3 (16.0)	88.2 (15.8)	91.5 (15.9)	90.4 (16.2)	0.270
**Impaired lung function** ^(a)^					
Impaired FEV1	85 (21.1%)	27 (28.1%)	30 (18.6%)	28 (19.2%)	0.153
Impaired FVC	105 (26.1%)	33 (34.4%)	33 (20.5%)	39 (26.7%)	0.048
Impaired FEV1 and/or FVC	116 (28.8%)	37 (38.5%)	39 (24.2%)	40 (27.4%)	0.044
**Types of ventilatory patterns** ^(b)^					
Normal spirometry	287 (71.2%)	59 (61.5%)	122 (75.8%)	106 (72.6%)	0.044
Spirometric abnormality pattern					0.906
Obstructive	50 (12.4%)	17 (17.7%)	16 (9.9%)	17 (11.6%)	
Non-obstructive	66 (16.4%)	20 (20.8%)	23 (14.3%)	23 (15.8%)	

(*) Adherence to the Mediterranean Diet was assessed using the 14-item MEDAS validated questionnaire that considers low (0–6 points), medium (7–8 points) and high (9–14 points) adherence [19]. Data are presented as number of patients (and percentage), and the mean (standard deviation) or as median [quartile 1; quartile 3], according to the type of variable. FEV1: force expiratory volume in the first second; FVC: forced vital capacity. According to ATS/ERS recommendations [20]: ^(^^a)^ Impaired lung function forced vital capacity (FVC) and/or forced expiratory volume in 1 s (FEV1) was <80% of the predicted value. ^(^^b)^ Normal spirometry (FVC y FEV1 ≥ 80% of the predicted value and FEV1/FVC ≥ 70%), obstructive ventilatory disorder (FEV1/FVC < 70%) and non-obstructive ventilatory disorder (FVC < 80% of the predicted value and FEV1/FVC > 70%).

**Table 3 nutrients-15-01272-t003:** Relationship between Mediterranean Diet Adherence Group and impaired pulmonary function (unadjusted, age/sex-adjusted and multivariable-adjusted logistic regression model).

	Impaired FEV1		Impaired FVC		Impaired FEV1 and/or FVC	
	OR (95% CI)	*p* Value	OR (95% CI)	*p* Value	OR (95% CI)	*p* Value
**Unadjusted model**						
Mediterranean Diet Adherence *						
low	reference		reference		reference	
medium	0.585 (0.322–1.062)	0.078	0.492 (0.279–0.869)	0.015	0.510 (0.416–0.946)	0.016
high	0.606 (0.331–1.112)	0.106	0.696 (0.398–1.216)	0.203	0.602 (0.348–1.042)	0.070
**Age/sex adjusted model**						
Mediterranean Diet Adherence *						
low	reference		reference		reference	
medium	0.565 (0.309–1.031)	0.063	0.465 (0.261–0.831)	0.010	0.477 (0.273–0.835)	0.009
high	0.575 (0.311–1.063)	0.078	0.648 (0.366–1.146)	0.136	0.550 (0.313–0.968)	0.038
Sex (women)	0.644 (0.392–1.057)	0.082	0.572 (0.359–0.912)	0.019	0.554 (0.351–0.872)	0.011
Age (years)	1.019 (0.994–1.043)	0.134	1.025 (1.002–1.048)	0.135	1.029 (1.006–1.052)	0.013
**Multivariable adjusted model**						
Mediterranean Diet Adherence *						
low	reference		reference		reference	
medium	0.582 (0.316–1.074)	0.083	0.475 (0.265–0.851)	0.012	0.487 (0.277–0.856)	0.012
high	0.590 (0.317–1.097)	0.095	0.643 (0.363–1.141)	0.131	0.556 (0.315–0.979)	0.042
Sex (women)	0.648 (0.390–1.078)	0.095	0.587 (0.366–0.940)	0.027	0.565 (0.356–0.896)	0.015
Age (years)	1.017 (0.993–1.042)	0.172	1.023 (1.000–1.047)	0.048	1.027 (1.004–1.050)	0.019
BMI (kg/m^2^)	1.002 (0.954–1.051)	0.950	1.020 (0.976–1.067)	0.383	1.011 (0.968–1.056)	0.616
Smoking current consumption						
Daily consumption (min, 10] cigarettes	reference		reference		reference	
Daily consumption (10, 20] cigarettes	0.764 (0.449–1.301)	0.322	0.900 (0.552–1.469)	0.674	0.886 (0.548–1.432)	0.621
Daily consumption (20, max] cigarettes	2.051 (0.936–4.495)	0.073	1.044 (0.461–2.365)	0.917	1.532 (0.709–3.310)	0.278

Data shown are logistic regression model as odds ratio (OR) with 95% confidence intervals (CI) and *p*-value. Analyses were Mediterranean Diet Adherence unadjusted, sex (man reference) and age (continuous years) adjusted, and adjusted by multiple variables: civil status, social class, comorbidities, body mass index (BMI; continuous, kg/m^2^), physical activity (IPAQ), alcohol intake (standard drink unit, SDU/week; an SDU equals 10 g of alcohol) and smoking current consumption (daily consumption categorised: 0–10 cigarettes, 11–20 cigarettes and more than 20 cigarettes). (*) Mediterranean Diet Adherence categorised: low 0–6, intermediate 7–8, and high 9–14 points, in MEDEAS questionnaire [19]); FEV1: force expiratory volume in the first second; FVC: forced vital capacity.

**Table 4 nutrients-15-01272-t004:** Relationship between Mediterranean Diet Adherence Group and impaired FEV1% and impaired FVC% predicted value.

	FEV1% Predicted Value		FVC% Predicted Value	
	β Regression Coefficient (95% CI)	*p* Value	β Regression Coefficient (95% CI)	*p* Value
**Unadjusted Model**				
Mediterranean Diet Adherence *				
low	reference		reference	
medium	2.906 (−1.507; 7.319)	0.196	3.332 (−0.716; 7.379)	0.106
high	1.873 (−2.623; 6.370)	0.413	2.239 (−1.885; 6.364)	0.286
**Age/sex adjusted model**				
Mediterranean Diet Adherence *				
low	reference		reference	
medium	3.044 (−1.139; 7.427)	0.173	3.629 (−0.341; 7.599)	0.073
high	2.005 (−2.474; 6.484)	0.379	2.651 (−1.407; 6.708)	0.200
Sex (women)	5.142 (1.566; 8.718)	0.005	6.029 (2.789; 9.268)	<0.001
Age (years)	−0.050 (−0.218; 0.119)	0.564	−0.165 (−0.317; −0.012)	0.035
**Multivariable adjusted model**				
Mediterranean Diet Adherence *				
low	reference		reference	
medium	3.203 (−1.133; 7.540)	0.145	3.660 (−0.285; 7.605)	0.069
high	1.481 (−2.955; 5.918)	0.487	2.239 (−1.786; 6.264)	0.275
Sex (women)	5.870 (2.212; 9.529)	0.002	6.220 (2.893; 9.546)	0.000
Smoking cumulative consumption (pack-year)	−0.142 (−0.272; −0.038)	0.008	−0.143 (−0.239; −0.048)	0.003
BMI (Kg/m^2^)		0.975	−0.193 (−0.494; 0.109)	0.209
Alcohol intake (SDU/week)	0.374 (0.074; 0.675)	0.013	0.244 (−0.028; 0.517)	0.079

Data shown are lineal regression model as β regression coefficient with 95% confidence intervals (CI) and *p*-value. Analyses were Mediterranean Diet Adherence unadjusted, sex (man reference) and age (continuous years) adjusted, and adjusted by multiple variables: civil status, social class, comorbidities, body mass index (BMI; continuous, kg/m^2^), physical activity (IPAQ), alcohol intake (standard drink unit, SDU/week; an SDU equals 10 g of alcohol) and smoking cumulative consumption (continuous, pack-year). (*) Mediterranean Diet Adherence categorised: low 0–6, intermediate 7–8, and high 9–14 points, in MEDEAS questionnaire [19]); FEV1: force expiratory volume in the first second; FVC: forced vital capacity.

## Data Availability

All data are considered confidential and treated according to Regulation 2016/679 of the European Parliament and Council of 27 April 2016 on Data Protection and Organic Law 3/2018, of 5 December, on the protection of personal data and guarantee of digital rights. Access is restricted to the research team by password. Data are available on reasonable request. The full dataset and statistical code are available from the corresponding (fmartin.tgn.ics@gencat.cat).

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
