# Peer review of "Mediterranean Diet and Lung Function in Adults Current Smokers: A Cross-Sectional Analysis in the MEDISTAR Project"

_nutrients, 2023, doi:10.3390/nu15051272_

Round 1

Reviewer 1 Report

Review of “Mediterranean diet and lung function in adult current smokers: A cross-sectional analysis in the MEDISTAR project” by Roxana-Elena Catalin et al.

This is an interesting study of how Mediterranean diet could influence lung function. The authors provided abundant information through an observational study conducted with 403 middle-aged patients that treated at 20 centers of primary care. The conclusion that Mediterranean diet may protect lung function may be interested by many researchers who study lung diseases. However, I struggled to get the major ideas of the analysis and therefore not entirely convinced by the conclusion. The authors should make immediate statement of each analysis and point out that the major/most important outcome that support this statement, rather than letting audience wonder among them.

Minor comments

1. Line 22, “some studies” could be “previous studies”

2. Line 36, “MeDi adherence seems to be inversely associated with the…” “seems” is a less confident word.

3. Figure 1 has a two group in the end but seems all analysis are done in three groups (Low, Medium, and High)?

4. Figure 2 did not provide useful information for the following analysis

Author Response

Please see the attachement!

Thank you!

Reviewer 2 Report

nutrients-2228485

The article is written in standard English and provides beneficial information on the current status of the Mediterranean diet and lung function in adult's current smok- 2ers.

The statistical processing of the data is presented at a high level. I suggest a violin plot representation for Figure 2 (will be clear and exciting, with fewer data).

References part: only 21 articles are from the last 5 years - I suggest, if possible, that any of the proposed articles be updated!

Author Response

Thank you!

Reviewer 3 Report

Dear authors,

the general opinion about this paper is: 

the paper uses sound scientific methodology, 

a proper survey sample and research methodology

the findings are presented in a good scientific manner and they posses added value for the existing literature and research in the area of smoking effects and health. 

The particular minor changes that should improve the paper quality refer to

methodology and introduction section:

The authors could with at least a sentence describe other 4 environmental factors because their text refers only to diet as a factor.

Line 46

Environmental factors, whereas lifestyle factors, including smoking and environmental exposure, physical activity and diet that are modifiable and can be changed

Line 148

Since in methodology these variables are environmental factors used for the criteria of participants’ selection give a wider explanation for environmental factor 1.)   high, moderate and low physical activity, and inactive

Then , 2), 3) blood pressure (systolic [SBP] and diastolic [DBP], in 150

Then 4)

for all other criteria used in methodology  5) blood pressure mmHg), height (cm, using a conventional stadiometer), weight

then 6) because blood sugar, cholesterol and other problematic high – increased levels of certain blood components have general and particular negative effects on human health and cause several long term or life long metabolic and other serious diseases

In methodology could you describe the criteria for alcohol use – low, medium high or moderate?

And describe what means low- medium – high psysical activity

The authors should revise the references according to the mdpi style

Take care: after number 1 no line but a full stop

authors fonts are not bold, but ordinary

after authors name a coma should be put…

between names a full stop

DOI with small letters

Italics for the book – journal title … etc.

See your refrences: 1- Singh D, Agusti A, Anzueto A, Barnes PJ, Bourbeau J, Celli BR, et al. Global Strategy for the Diagnosis, Management, and 618 Prevention of Chronic Obstructive Lung Disease: The GOLD science committee report 2019. Eur Respir J. 2019;53(5):1900164. 619 DOI: 10.1183/13993003.00164-2019. 620

2. 2- Reddel HK, Bacharier LB, Bateman

best regards, 

the reviewer

Author Response

Thank you!

Round 2

Reviewer 1 Report

The authors have addressed all my questions and concerns.